# Long-term consequences of benzodiazepine-induced neurological dysfunction: A survey

**Alexis D. Ritvo**[1]⊛*, **D. E. Foster**[2]⊛, **Christy Huff**[3]⊛, **A. J. Reid Finlayson**[4]⊛, **Bernard Silvernail**[5]⊛, **Peter R. Martin**[6]⊛

**1** Department of Psychiatry, University of Colorado School of Medicine, Aurora, Colorado, United States of America, **2** Benzodiazepine Action Work Group, Colorado Consortium for Prescription Drug Abuse Prevention, Aurora, Colorado, United States of America, **3** Benzodiazepine Information Coalition, Midvale, Utah, United States of America, **4** Department of Psychiatry and Behavioral Sciences, Vanderbilt University Medical Center, Nashville, Tennessee, United States of America, **5** Alliance for Benzodiazepine Best Practices, Portland, Oregon, United States of America, **6** Department of Psychiatry and Behavioral Sciences and Department of Pharmacology, Vanderbilt University Medical Center, Nashville, Tennessee, United States of America

⊛ These authors contributed equally to this work.
* ALEXIS.RITVO@cuanschutz.edu

## Abstract

### Background

Acute benzodiazepine withdrawal has been described, but literature regarding the benzodiazepine-induced neurological injury that may result in enduring symptoms and life consequences is scant.

### Objective

We conducted an internet survey of current and former benzodiazepine users and asked about their symptoms and adverse life events attributed to benzodiazepine use.

### Methods

This is a secondary analysis of the largest survey ever conducted with 1,207 benzodiazepine users from benzodiazepine support groups and health/wellness sites who completed the survey. Respondents included those still taking benzodiazepines (n = 136), tapering (n = 294), or fully discontinued (n = 763).

### Results

The survey asked about 23 specific symptoms and more than half of the respondents who experienced low energy, distractedness, memory loss, nervousness, anxiety, and other symptoms stated that these symptoms lasted a year or longer. These symptoms were often reported as *de novo* and distinct from the symptoms for which the benzodiazepines were originally prescribed. A subset of respondents stated that symptoms persisted even after benzodiazepines had been discontinued for a year or more. Adverse life consequences were reported by many respondents as well.

**Data Availability Statement:** The data are held in a public repository: https://osf.io/cewgb/ DOI 10.17605/OSF.IO/CEWGB

**Funding:** The authors received no specific funding for this work.

**Competing interests:** I have read the journal's policy and the authors of this manuscript have the following competing interests: Alexis D. Ritvo is contracted as the medical director for the national non-profit the Alliance for Benzodiazepine Best Practices. Alexis D. Ritvo and D. E. Foster are co-chairs for the Benzodiazepine Action Work Group with the Colorado Consortium for Prescription Drug Abuse Prevention. D. E. Foster is also the founder and owner of Easing Anxiety. Christy Huff is a director with the Benzodiazepine Information Coalition. Bernie Silvernail Sanders is president of the Alliance for Benzodiazepine Best Practices. This does not alter our adherence to PLOS ONE policies on sharing data and material.

## Limitations

This was a self-selected internet survey with no control group. No independent psychiatric diagnoses could be made in participants.

## Conclusions

Many prolonged symptoms subsequent to benzodiazepine use and discontinuation (benzodiazepine-induced neurological dysfunction) have been shown in a large survey of benzodiazepine users. Benzodiazepine-induced neurological dysfunction (BIND) has been proposed as a term to describe symptoms and associated adverse life consequences that may emerge during benzodiazepine use, tapering, and continue after benzodiazepine discontinuation. Not all people who take benzodiazepines will develop BIND and risk factors for BIND remain to be elucidated. Further pathogenic and clinical study of BIND is needed.

## Introduction

Acute benzodiazepine withdrawal and its effective treatment are well known and have been described in the literature [1–4]. However, symptoms that persisted for months or even years after complete benzodiazepine discontinuation were observed decades ago [5, 6]. Prior to our survey, the largest study of this phenomenon, in which 50 subjects were examined, was carried out in 1987 and noted that symptoms in some patients persisted for months to years [7, 8]. Since then, clinical recognition of this condition, treatment strategies to address it, and a fundamental mechanistic understanding of how it differs from acute withdrawal remain confounded.

The nature of protracted withdrawal symptoms leads to a variety of interpretations, including the commonly held belief that they merely represent the return of the original symptoms for which the benzodiazepines were originally prescribed. However, if symptoms appeared *de novo* during and after benzodiazepine cessation, they may be attributed to a different or unrelated cause than that for which a benzodiazepine was actually prescribed. These protracted symptoms and other sequelae associated with the use, tapering, and discontinuation of benzodiazepines may be a distinct clinical entity.

The lack of descriptive nomenclature for enduring symptoms associated with benzodiazepine use limits both the clinical identification of this condition and informed discussion of risk with patients. Inadequate terminology such as "withdrawal," "subacute withdrawal," "protracted withdrawal," "post-acute withdrawal syndrome" (PAWS), rebound, and other terms without clear definitions appear in the scant literature about prolonged symptoms after benzodiazepine discontinuation. The focus on specific symptoms and in comparison to acute withdrawal symptoms from other substances, such as alcohol or opioids, implies that benzodiazepine withdrawal follows a well-defined acute trajectory which resolves over a relatively short period of time. These findings and the results of our earlier reports [9, 10] conflict with some of the literature [11].

To the best of our knowledge, this online survey is the largest ever conducted among benzodiazepine users. Its objective was to better describe and quantify the life consequences associated with these prolonged symptoms. It described constellations of benzodiazepine-induced and sometimes *de novo* symptoms, many of which lasted beyond a year and which were often accompanied by adverse life consequences.

Our objective was to better describe and quantify the life consequences associated with these prolonged symptoms.

## Methods and materials

This study represents a secondary analysis of the results from an internet survey published previously [9]. It was approved by the Vanderbilt University Institutional Review Board (IRB) #20052, and did not require written informed consent because it was conducted as an anonymous survey that began with a question which recorded each respondent's consent for participation.

A medical statistician produced the initial results of this survey utilizing SAS Software. Subsequent data analysis was performed in greater detail by an experienced data scientist who imported the survey data into a custom SQL Server data model. Customized queries were employed to obtain correlations among the data. In particular, this analysis examined conditions for which benzodiazepines were prescribed and compared them to protracted symptoms reported by patients who were tapering or had discontinued benzodiazepine use. Adverse life consequences experienced by benzodiazepine patients, as reported in the survey, were also correlated to protracted symptoms. The complete survey form appears in S1 Appendix. The questions and multiple-choice answers used in the survey were derived from a subset of a longer list of benzodiazepine-associated symptoms report by Ashton [12] and Wright [1].

All analyses were delivered via a structured reporting process and validated against the original SAS reports. The survey was made available online through websites and internet benzodiazepine support groups and general health and wellness groups.

## Results

A total of 1,207 respondents finished the survey although not all respondents gave an answer to every question and some questions allowed for multiple answers. Respondents to the survey might have been taking their full dose of benzodiazepines, engaged in the process of tapering off benzodiazepines, or had fully discontinued benzodiazepines. Respondents were asked to select among 23 symptoms they may have experienced and to indicate the duration of each symptom (see S1 Appendix). Of all respondents, 88.1% reported having anxiety, nervousness, or fear; 86.9% sleep disturbances; 86.2% low energy levels; and 85.3% difficulty focusing or distractedness. Some respondents reported these symptoms occurring following complete cessation of benzodiazepines and for long-term durations of months or

**Table 1. Of those who reported the following symptoms shown in the table, over half of respondents stated the symptom lasted $\geq$ 1 year.**

| Symptom | Symptom persisted $\geq$ 1 year |
|---|---|
| Low energy | 59.9% |
| Difficulty focusing, distractedness | 58.3% |
| Memory loss | 57.5% |
| Nervous, anxiety | 57.0% |
| Sleep disturbances | 56.4% |
| Sensitivity to sights and sounds | 54.3% |
| Digestive issues | 52.2% |
| Symptoms triggered by food or drink | 52.0% |
| Muscle weakness | 51.2% |
| Body aches and pains | 50.7% |

years. In fact, 76.6% of all affirmative answers on symptom questions reported symptom durations to be months or "one year or longer." The most frequently reported symptoms lasting one year or more appear in Table 1.

The symptoms reported in Table 1 occurred across all respondents, regardless of taper status and cause for original prescription of the benzodiazepine. When groups were separated into those still taking benzodiazepines at full dose (11.3%), those tapering (24.4%), and those who had completely discontinued benzodiazepines (63.2%), results showed that respondents who were taking the full-dose at the time of the survey reported experiencing the fewest symptoms, with modest differences between tapering and discontinued respondents. The survey queried respondents about the conditions or situation for which the benzodiazepines were prescribed. The most common reasons for prescriptions were situational anxiety (43.7%), insomnia (40.3%), panic attacks (39.9%), depression (33.0%), and generalized anxiety disorder (23.7%). However, the prolonged symptoms after benzodiazepine use, tapering, or cessation frequently did not match the reason for which the benzodiazepines were originally prescribed. See Table 2.

More than half of all respondents (54.7%) experienced 17 or more symptoms of the 23 listed; and over 40% of these stated the symptoms as lasting "one year or longer."

In addition to enduring symptoms associated with benzodiazepines, many respondents reported that adverse consequences had occurred in multiple areas of their life (see Table 3). Over 90% of respondents attributed one or more general adverse life consequences to benzodiazepine use. A large majority of respondents (79.3%) reported six to 13 general life consequences, and 53.2% of respondents reported eight or more specific life consequences, all of which they attributed to benzodiazepine use. On average, each respondent had 8.1 of the 16 adverse life consequences. Over 90% of respondents attributed one or more general adverse life consequences to benzodiazepine use. These included adverse effect on work life, fun and recreation, ability to take care of home and other, ability to drive or walk, social interactions or friendships, and relationships with spouse or family. More specific adverse life consequences were also reported (see Table 3) and were associated with a higher average frequency of symptoms than the overall survey population, 19/23 versus 15/23 symptoms, respectively. A subpopulation of respondents (n = 225, 18.6%) stated that none of these specific negative life consequences applied to them and, on average, reported their symptom duration in days or weeks rather than months or years; in other words, they experienced acute withdrawal symptoms.

Those respondents taking a full dose of benzodiazepine tended to the lowest rates of adverse life consequences. See Table 4.

A total of 763 respondents reported they had discontinued benzodiazepines, of whom 426 stated they had been off benzodiazepines for a year or more. Adverse life consequences

**Table 2. Proportion of respondents who experienced a protracted symptom for which the benzodiazepines were not originally prescribed.**

| Reason for the original benzodiazepine prescription | Proportion of respondents (n = 1,207) who reported this symptom but were not prescribed for it |
| --- | --- |
| Situational anxiety/anxiety | 55.6% |
| Insomnia | 57.5% |
| Digestive, stomach/gut issues | 75.8% |
| Restlessness | 95.3% |
| Muscle spasms | 88.8% |
| Pain, nerve spasms | 88.1% |

**Table 3. Specific life consequences correlated to symptoms attributed to benzodiazepine use.** A total of 23 symptoms could be selected in the survey. For all life consequences, the average duration of reported symptoms was >1 year.

| Specific Adverse life consequences | Total reporting (% of total) | Average number of symptoms in this group |
|---|---|---|
| Significantly affected marriage, other relationships | 686 (56.8%) | 18.2 |
| Suicidal thoughts or attempted suicide | 657 (54.4%) | 18.3 |
| Lost a job, fired, became unable to work | 585 (46.8%) | 18.5 |
| Experienced significant increase in medical costs | 494 (40.9%) | 18.5 |
| Loss of wages or lower wages in a reduced job capacity | 394 (32.6%) | 18.4 |
| Lost savings or retirement funds | 322 (26.7%) | 19.1 |
| Violent thoughts or actual violence against others | 284 (23.5%) | 19.3 |
| Lost a home | 152 (12.6%) | 19.2 |
| Lost a business, if business owner | 101 (8.4%) | 18.4 |
| Lost child custody | 31 (2.6%) | 20.9 |

reported by those who had discontinued benzodiazepines for a year or more were deemed severe or worse by 55.9% to 83.6% of respondents. See Table 5.

## Discussion

This analysis presents survey evidence that enduring symptoms along with adverse life consequences emerged *de novo* with benzodiazepine use. Although protracted symptoms following discontinuation of benzodiazepine use have been reported previously [9, 13, 14], it has generally been tacitly assumed that these symptoms were withdrawal phenomena that would resolve with time. This study reveals something entirely different: that new, and often persistent, symptoms induced by the use of benzodiazepines may emerge while using, tapering, or after discontinuing these medications. In fact, a subset of respondents who had completely discontinued benzodiazepines, including those who had ceased taking benzodiazepines for a year or more, continued to experience enduring life consequences.

This analysis showed ≥ 17 symptoms of ≥ 1 year's duration post-discontinuation were reported by over 40% of respondents. This is not the first report that benzodiazepine "withdrawal" symptoms persist long after drug discontinuation. As far back as 1981, Hallström and Lader found elevated Hamilton anxiety scores several months after patients had withdrawn from benzodiazepines [5]. Smith and Wesson observed that symptoms following withdrawal from low-dose benzodiazepines typically took six to 12 months to subside completely [6]. In 1987, Ashton, whose study of 50 patients had been to our knowledge the largest study of prolonged benzodiazepine sequelae before our survey, noted symptoms lasting more than a year post-withdrawal [7]. Ashton also wrote that ". . .all patients had a variety of anxiety/depressive symptoms on presentation, and these had been gradually increased over the years despite continuous benzodiazepine use" [7]. A four-week, double-blind, placebo-controlled diazepam withdrawal study also showed elevated post-withdrawal symptoms [15]. Eight weeks after the end of withdrawal, mean scores for headache, dizziness, depression, tinnitus, paresthesia, and motor symptoms remained higher than pre-withdrawal scores; other symptoms had declined but few had disappeared [15]. A case series (n = 104) is discussed as part of the unpublished report that precipitated the 2020 benzodiazepine-class FDA boxed warning. In this report, of

**Table 4. Adverse life consequences of those on full dose benzodiazepine therapy, those tapering, and those who had completely discontinued benzodiazepines.** Totals represent the number of respondents who answered this question in the affirmative and the percentages indicate the proportion of the specific population who reported those consequences.

| Life consequences | Total (%) n = 1,207 | Full dose n = 136 | Tapering n = 294 | Discontinued n = 763 |
|---|---|---|---|---|
| *To what extent has your condition affected your work or personal life? How severely did this problem affect you? (Respondents could answer on a scale of 1 to 6, where 1 was "not at all" and 6 was "enormous problem.") Response rates are for those who stated ≥ 2.* | | | | |
| Work life | 1000 (82.9%) | 90 (66.2%) | 258 (87.8%) | 650 (85.2%) |
| Fun, recreation, hobbies | 1072 (88.8%) | 98 (72.1%) | 280 (95.2%) | 692 (90.7%) |
| Ability to care for home, others | 1031 (85.4%) | 91 (66.9%) | 271 (92.2%) | 667 (87.4%) |
| Ability to drive or walk | 921 (76.3%) | 77 (56.6%) | 233 (79.3%) | 610 (79.9%) |
| Social interaction, friendships | 1042 (86.3%) | 92 (67.6%) | 275 (93.5%) | 673 (88.2%) |
| Relationships with spouse, family | 1023 (84.8%) | 88 (64.7%) | 272 (92.5%) | 661 (86.6%) |
| *Specifically, have any of these been consequences of your benzodiazepine use or withdrawal?* | | | | |
| Significantly affected marriage, other relationships | 686 (56.8%) | 63 (46.3%) | 165 (56.1%) | 456 (59.8%) |
| Suicidal thoughts or attempted suicide | 657 (54.4%) | 50 (36.8%) | 176 (59.9%) | 430 (56.4%) |
| Lost a job, fired, became unable to work | 565 (46.8%) | 52 (38.2%) | 147 (50.0%) | 365 (47.8%) |
| Experienced significant increase in medical costs | 494 (40.9%) | 39 (28.7%) | 134 (45.6%) | 320 (41.9%) |
| Loss of wages or lower wages in reduced job capacity | 394 (32.6%) | 31 (22.8%) | 97 (33.0%) | 265 (34.7%) |
| Lost savings or retirement funds | 322 (26.7%) | 19 (14.0%) | 78 (26.5%) | 223 (29.2%) |
| Violent thoughts or actual violence against others | 284 (23.5%) | 24 (17.6%) | 76 (25.9%) | 184 (24.1%) |
| Lost a home | 152 (12.6%) | 13 (9.6%) | 39 (13.3%) | 99 (13.0%) |
| Lost a business, if business owner | 101 (8.4%) | 11 (8.1%) | 24 (8.2%) | 65 (8.5%) |
| Lost child custody | 31 (2.6%) | 5 (3.7%) | 5 (1.7%) | 21 (2.8%) |

Note that there were 1,207 respondents but only 1,193 respondents answered these questions.

the patients who reported withdrawal, the mean duration of withdrawal was 9.5 months [16]. Prolonged symptoms after benzodiazepine discontinuation have been reported elsewhere, ranging from anxiety, insomnia, nightmares, and deficits in memory or concentration [17]. While few formal studies have examined enduring benzodiazepine symptoms, there are thousands of accounts online from individuals who report prolonged and distressing symptoms even after complete drug discontinuation [18].

**Table 5. Respondents who had completely discontinued benzodiazepines for at least one year at the time of the survey (n = 426) rated the severity of life consequences on a scale of 1 to 6, with 6 the most severe.**

| Life Consequences | Not at all a problem, mild problem, or moderate problem (1, 2, 3) | Severe, quite severe, or enormous problem (4, 5, 6) |
|---|---|---|
| *To what extent has your condition affected your work or personal life? How severely did this problem affect you? (Respondents could answer on a scale of 1 to 6, where 1 was "not at all" and 6 was "enormous problem.") Response rates are for those who stated ≥ 2.* | | |
| Fun, recreation, hobbies | 70 (16.4%) | 356 (83.6%) |
| Work life | 88 (20.7%) | 338 (79.3%) |
| Social interaction, friendships | 99 (23.2%) | 327 (76.8%) |
| Ability to take care of home, others | 117 (27.5%) | 309 (72.5%) |
| Relationships with spouse, family | 133 (31.2%) | 293 (68.8%) |
| Ability to drive or walk | 188 (44.1%) | 238 (55.9%) |

The occurrence of adverse life consequences associated with benzodiazepines has not been thoroughly studied. Although efforts were made, statistical correlations between specific life consequences reported in our study and symptoms could not be drawn, but it appears based on available data from the respondents in the survey that enduring symptoms may have played an important role in damaging life consequences they experienced. This study shows that over 80% of respondents identified more than five serious life consequences which they attributed to benzodiazepine use. To the best of our knowledge, this is the first study to explore adverse life consequences associated with these enduring symptoms, of which many were neurocognitive. A meta-analysis of cognitive effects found that long-term benzodiazepine users were more impaired in all cognitive categories than the controls [19, 20]. This supports our findings, because several life consequences reported in our survey are likely related to impaired cognitive functioning. This would align with recently published findings from Europe where neuropsychological evaluation of cognition in 92 long-term benzodiazepine patients found 20.7% could be categorized as having cognitive impairment across all domains, with processing speed and sustained attention the worst-performing domains [21].

The term benzodiazepine-induced neurological dysfunction and its acronym BIND was coined as an effort by a separately convened work group of experts to provide a name for this condition that may serve both clinicians and the patients who suffer from this condition. See S2 Appendix. BIND serves as a clinically serviceable name for the enduring neurological sequelae of benzodiazepine use and would reify this condition for healthcare professionals. Patients in our survey sometimes wrote in comments that they felt like healthcare professionals did not take them seriously or frankly challenged or disbelieved their long-lasting symptoms [9]. Recognizing this condition with a specific medically accepted term may encourage more professional compassion, better treatment, and future research. Any medical condition with many vague or overlapping names or without a name can too easily be misdiagnosed or dismissed as insignificant or nonexistent. A name would reify this clinical entity. Practical, evidence-based, safe and effective approaches are urgently needed for benzodiazepine deprescribing and managing the enduring neurological sequelae of benzodiazepine use. The name BIND is an important first step in this direction. Thus, BIND describes a constellation of functionally limiting neurologic symptoms (both physical and psychological) that are the consequence of neuroadaptation and/or neurotoxicty resulting from benzodiazepine exposure.

BIND also includes disturbing life consequences. Unlike other reports about benzodiazepine use and discontinuation, our survey took into account both symptoms and adverse life events, such as financial loss, termination of employment, and other devasting events. The subset of patients who used benzodiazepines and developed BIND experience a bewildering, sometimes severe, set of prolonged effects that have gone largely unrecognized by the medical profession [22]. The mechanisms underlying these prolonged effects have not been elucidated but are likely different from the mechanisms of acute withdrawal, which are well understood [1].

There are only a few studies of low or very low quality evaluating the safety and effectiveness of pharmacological interventions to help manage the symptoms of chronic benzodiazepine use and none of these interventions have been shown consistently to be effective across significant portions of those affected [23]. Since benzodiazepine users are a heterogeneous population, it is unlikely there is a one-size-fits-all approach to tapering and discontinuation [24].

Over 30 million Americans report past-year use of benzodiazepines and this population is heterogenous. It includes old and young, fit and frail, and all demographic groups. Many of these benzodiazepine users are at elevated risk for BIND, which may go undiagnosed. Even when BIND is diagnosed, treatment protocols are lacking. While most benzodiazepine users do not develop BIND, the risk factors for BIND are not known. Since benzodiazepines are

among the most frequently prescribed drugs in the United States, treatments for BIND represent an urgent unmet medical need [25]. This warrants greater and more in-depth research.

Our survey was concluded prior to the outbreak of the pandemic, and it is not known how the lockdowns and COVID-19 affected substance use disorders in general or the use of benzodiazepines in particular [26]. According, there is also limited research on how the pandemic impacted benzodiazepine use patterns and the effect that the emergence of so-called "designer benzodiazepines" have had. This is a very complex topic because data on use must be disentangled from prevailing trends and tendencies in drug use patterns that may have been unrelated to the pandemic.

A growing concern is that individuals being tapered or deprescribed benzodiazepines too abruptly might turn to what is available to them, including alcohol, opioids, central nervous system depressants, and, of increasing concern are "designer benzodiazepines," such as diclazepam, conazolam, and nitrazolam; phenazepam and etizolam have been licensed as medical agents in some countries but not in the United States or Western Europe [27]. Over two dozen distinct "designer benzodiazepines" have been identified and many can be purchased online [28]. From the very limited available evidence, it appears that the use of designer benzodiazepines typically occurs in polysubstance abuse rather than in physician-prescribed benzodiazepine prescribing [29]. Our survey did not ask about these products; however, 90.4% of our respondents reported they definitely or mostly took benzodiazepines as prescribed.

Our study has several limitations. It is based on a self-selected group of respondents who were recruited primarily through benzodiazepine support group websites and may not be representative of all benzodiazepine users. There was no control group. It was a multiple-choice survey and although write-in information was accepted, our results are based entirely on responses from multiple-choice questions. The survey was anonymous and there was no access to the respondents' medical records to confirm their benzodiazepine use or status. The symptoms included in the multiple-choice survey were a subset of symptoms provided by Ashton [12] and Wright [1], and does not necessarily reflect the complete range of symptoms experienced by respondents. Since this was a survey, we were unable to determine whether respondents met criteria for a formal psychiatric disorder that contains the symptoms. No exclusion criteria were used for old age, comorbidities, or substance use disorder. Respondents were not asked if they were taking or tapering from other sedating or non-benzodiazepine hypnotic drugs. No questions were asked that might have allowed baseline symptoms to be compared with symptoms at other points in the trajectory of benzodiazepine use.

## Conclusions

While acute benzodiazepine withdrawal is well described in the literature, there is far less known about the often distressing and enduring symptoms which impair life functioning in those who have discontinued or are in the process of discontinuing benzodiazepines. We propose the term benzodiazepine-induced neurological dysfunction (BIND) for this constellation of symptoms. Our survey shows that for some benzodiazepine users, these symptoms are severe, life altering, and not infrequent. A significant subpopulation of respondents with BIND reported multiple and severe symptoms, many of which were not the symptoms for which the benzodiazepines were originally prescribed. The mechanisms of BIND, its clinical course, risk factors, and treatment modalities warrant further study.

## Supporting information

**S1 Appendix. Is the survey in its entirety.**
(DOCX)

**S2 Appendix. Describes the efforts of the benzodiazepine nosology workgroup.**
(DOCX)

## Acknowledgments

The authors gratefully acknowledge the work of Dr. Jane Macoubrie who was instrumental in creating the original survey, envisioning this publication, and supporting efforts at all levels to better explore the nature of these symptoms. The authors extend heartfelt thanks to all of the respondents who shared their experiences in the survey.

The authors gratefully acknowledge the support, voting, and vigorous debate provided by the Benzodiazepine Nosology Workgroup. The group consists of the authors plus, in alphabetical order: Sumit Agarwal, MD, Harvard Medical School and Brigham and Women's Hospital, Boston, Massachusetts USA; Richard Bailey, BSc (Hons), Guy's and St Thomas' NHS Foundation Trust, London UK; Christopher Blazes, MD, Oregon Health Sciences University and Veterans Administration Medical Center, both in Portland, Oregon USA; Leslie Brooks, MD, Sunrise Community Health and North Colorado Health Alliance, Evans, Colorado USA; Jaden Brandt, Msc.Pharm, University of Manitoba College of Pharmacy, Winnipeg, Manitoba Canada; Cathal Cadogon, PhD, School of Pharmacy and Pharmaceutical Sciences, Trinity College Dublin, Ireland; Doryn Davis Chervin, DrPH, Chervin and Associates, Cherry Hill, New Jersey USA; David Crabtree, MD, QuitGenius and PlushCare, San Francisco, California USA; Tim MacDonald, MD, Griffith University and Currumbin Clinic, Currumbin, Queensland Australia; Darrin Mangiacarne, DO, MPH, DFASAM, FAOAAM, Banyan Treatment Centers, Indianapolis, Indiana USA; Lori Mor, PharmD, Prisma Health Midlands, Family Medicine Residency Program, Florida, USA; Chinyere Ogbonna, MD, MPH, Kaiser Permanente, San Jose and Stanford Health, Stanford, both in California USA; Jocelyn Pederson; Arwen Podesta, MD, Tulane University School of Medicine, New Orleans, Louisiana USA; Erick Turner, MD, Department of Psychiatry, Oregon Health & Science University, Portland, Oregon USA; Jayne Violette, PhD, University of South Carolina Beaufort, Bluffton, South Carolina USA; and Steven Wright, MD [retired].

The authors acknowledge Jo Ann LeQuang for medical writing services, which were covered by the Alliance for Benzodiazepine Best Practices.

The authors acknowledge the support of the Alliance for Benzodiazepine Best Practices for implementing the discussions that led to the creation of this article.

## Author Contributions

**Conceptualization:** Alexis D. Ritvo, D. E. Foster, A. J. Reid Finlayson, Bernard Silvernail, Peter R. Martin.

**Data curation:** D. E. Foster.

**Formal analysis:** D. E. Foster.

**Investigation:** Christy Huff.

**Methodology:** D. E. Foster.

**Project administration:** Christy Huff.

**Validation:** D. E. Foster.

**Writing – original draft:** Alexis D. Ritvo, D. E. Foster, Christy Huff, A. J. Reid Finlayson, Bernard Silvernail, Peter R. Martin.

**Writing – review & editing:** Alexis D. Ritvo, D. E. Foster, Christy Huff, A. J. Reid Finlayson, Bernard Silvernail, Peter R. Martin.

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
