## [Decision Letter · Decision Letter 0]

14 Mar 2023

PONE-D-23-04336Long-term Consequences of Benzodiazepine-Induced Neurological Dysfunction: A SurveyPLOS ONE

Dear Dr. Ritvo,

Thank you for submitting your manuscript to PLOS ONE. After careful consideration, we feel that it has merit but does not fully meet PLOS ONE’s publication criteria as it currently stands. Therefore, we invite you to submit a revised version of the manuscript that addresses the points raised during the review process.

We look forward to receiving your revised manuscript.

Kind regards,

Simona Zaami

Academic Editor

PLOS ONE

Journal Requirements:

"I have read the journal's policy and the authors of this manuscript have the following competing interests: Alexis Ritvo is contracted as the medical director for the national non-profit the Alliance for Benzodiazepine Best Practices. Alexis Ritvo and D Foster volunteer as co-chairs for the Benzodiazepine Action Work Group with the Colorado Consortium for Prescription Drug Abuse Prevention. D Foster is also the founder and owner of Easing Anxiety. Christy Huff is a director with the Benzodiazepine Information Coalition. Bernie Sanders is president of the Alliance for Benzodiazepine Best Practices. "

6. We note that you have referenced (unpublished on page 13) which has currently not yet been accepted for publication. Please remove this from your References and amend this to state in the body of your manuscript: (ie “Bewick et al. [Unpublished]”) as detailed online in our guide for authors

Additional Editor Comments:

Dear Authors,

Thanks for your submission to Plos One.

I am hereby requesting that you amend their manuscript according to both reviewers' comments and suggestions.

Best regards,

Prof. Simona Zaami

Reviewers' comments:

Reviewer's Responses to Questions

**Comments to the Author**

1. Is the manuscript technically sound, and do the data support the conclusions?

Reviewer #1: Yes

Reviewer #2: Yes

2. Has the statistical analysis been performed appropriately and rigorously? 

Reviewer #1: Yes

Reviewer #2: Yes

3. Have the authors made all data underlying the findings in their manuscript fully available?

Reviewer #1: Yes

Reviewer #2: Yes

4. Is the manuscript presented in an intelligible fashion and written in standard English?

Reviewer #1: Yes

Reviewer #2: Yes

5. Review Comments to the Author

Reviewer #1: It was my pleasure to review the manuscript Long-term Consequences of Benzodiazepine-Induced Neurological Dysfunction: A

Survey.

The article revolves around a sound and painstaking analysis of 1,207 benzodiazepine users from benzodiazepine support groups and health/wellness sites.

The article is quite informative and cogently enunciated in its most relevant findings, which are ultimately a worthy research contribution likely to appeal to a rather broad readership of mental health professionals and addiction specialists. Shedding a light on the underlying factors determining or contributing to neurological repercussions from BDZ use is essential for forensic medicine and public health as well, given the far-reaching implications thereof.

In that regard, I feel that the article does not go far enough, and does not fully succeed in making the most out of its data analysis.

More depth needs to be added to the Discussion, also mentioning "substitute" BDZ substances and the threat they pose in terms of detection and control, in addition to the psychiatric implications.

Thereference pool should be enhanced, consider the following:

PMID: 32144953.

PMID: 29543325

PMID: 31799633.

PMID: 36041417.

The article is clear and straightforward overall, and the tables are meaningful and well conceived.

Making it more comprehensive will add to its relevance and improve balance and development.

Sincerely.

Reviewer #2: Dear Authors,

I have read and mostly appreciated your article titled Long-term Consequences of Benzodiazepine-Induced Neurological Dysfunction: A

Survey, in which the distinctive features and complex dynamics at the heart of BIND have been expounded upon quite effectively and in a scientifically sound fashion.

The article has qualities and strengths which make it a praiseworthy scientific research contribution. It has considerable elements of novelty, relevance and thorougness as far as its stated objective is. The methodology is sound, as far as I could determine.

The one area in which the article falls short is the limited scope in terms of mentioning and elaborating on factors such as screening and detection and policies and measures aimed at mitigating the spread of BDZs with an eye on designer drugs, i.e. replacements to BDZs and other substances of abuse.

Briefly addressing such elements of discussion would certainly contribute to making the article more comprehensive and well-rounded, which would be advisable in light of the uniquely consequential issues arising fron BDZs abuse. It is also worth mentioning the impact of the COVID-19 pandemic on abuse dynamics overall. Too many sources are older than five years.

The following sources ought to be drawn upon and cited as well:

Zaami S, Graziano S, Tittarelli R, Beck R, Marinelli E. BDZs, Designer BDZs and Z-drugs: Pharmacology and Misuse Insights. Curr Pharm Des. 2022;28(15):1221-1229. doi: 10.2174/1381612827666210917145636.

Moosmann B, Auwärter V. Designer Benzodiazepines: Another Class of New Psychoactive Substances. Handb Exp Pharmacol. 2018;252:383-410. doi: 10.1007/164_2018_154.

Lo Faro AF, Venanzi B, Pilli G, Ripani U, Basile G, Pichini S, Busardò FP. Ultra-high-performance liquid chromatography-tandem mass spectrometry assay for quantifying THC, CBD and their metabolites in hair. Application to patients treated with medical cannabis. J Pharm Biomed Anal. 2022 Aug 5;217:114841. doi: 10.1016/j.jpba.2022.114841.

Negro F, Di Trana A, Marinelli S. The effects of the COVID-19 pandemic on the use of the performance-enhancing drugs. Acta Biomed. 2022 Jan 19;92(6):e2021401. doi: 10.23750/abm.v92i6.12377.

Mannocchi G, Di Trana A, Tini A, Zaami S, Gottardi M, Pichini S, Busardò FP. Development and validation of fast UHPLC-MS/MS screening method for 87 NPS and 32 other drugs of abuse in hair and nails: application to real cases. Anal Bioanal Chem. 2020 Aug;412(21):5125-5145. doi: 10.1007/s00216-020-02462-6.

Walton SE, Krotulski AJ, Logan BK. A Forward-Thinking Approach to Addressing the New Synthetic Opioid 2-Benzylbenzimidazole Nitazene Analogs by Liquid Chromatography-Tandem Quadrupole Mass Spectrometry (LC-QQQ-MS). J Anal Toxicol. 2022 Mar 21;46(3):221-231. doi: 10.1093/jat/bkab117.

Napoletano S, Basile G, Lo Faro AF, Negro F. New Psychoactive Substances and receding COVID-19 pandemic: really going back to "normal"? Acta Biomed. 2022 May 11;93(2):e2022186. doi: 10.23750/abm.v93i2.13008.

The article is overall well-written and coherently assembled. With a few adjustments, I believe it could make for a valuable and meaningful contribution to a highly relevant area of research.

Best regards.

6. PLOS authors have the option to publish the peer review history of their article (what does this mean?). If published, this will include your full peer review and any attached files.

Reviewer #1: No

Reviewer #2: No

---

## [Author Response · Author response to Decision Letter 0]

31 Mar 2023

Thank you for the opportunity to revise our manuscript. Please see attached rebuttal letter worksheet for our response to each point raised by the academic editor and reviewers.

---

## [Editor Report · Decision Letter 1]

27 Apr 2023

Long-term Consequences of Benzodiazepine-Induced Neurological Dysfunction: A Survey

PONE-D-23-04336R1

Dear Dr. Ritvo,

We’re pleased to inform you that your manuscript has been judged scientifically suitable for publication and will be formally accepted for publication once it meets all outstanding technical requirements.

Kind regards,

Simona Zaami

Academic Editor

PLOS ONE

Additional Editor Comments (optional):

Dear Authors,

I have gone over the latest version of the manuscript titled Long-term Consequences of Benzodiazepine-Induced Neurological Dysfunction: A Survey, it is my belief that you have mostly succeeded in improving the manuscript by addressing the reviewers' comments, and amend their article accordingly.

I feel that in light of the improvements made, the article is now more comprehensive and well-rounded overall.

It will make for a valuable contribution to a highly relevant field of toxicology research.

Best regards,

Prof. Simona Zaami

---

## [Editor Report · Acceptance letter]

21 Jun 2023

PONE-D-23-04336R1 

Long-term consequences of benzodiazepine-induced neurological dysfunction: A survey 

Dear Dr. Ritvo:

I'm pleased to inform you that your manuscript has been deemed suitable for publication in PLOS ONE. Congratulations! Your manuscript is now with our production department. 

Kind regards, 

on behalf of

Dr. Simona Zaami 

Academic Editor

PLOS ONE